DATA RELEASE

# Manual curation and phylogenetic analysis of chitinase family genes in the Asian citrus psyllid, *Diaphorina citri*

Teresa D. Shippy[1], Sherry Miller[1,2], Blessy Tamayo[3], Prashant S. Hosmani[4], Mirella Flores-Gonzalez[4], Lukas A. Mueller[4], Wayne B. Hunter[5], Susan J. Brown[1], Tom D'Elia[3] and Surya Saha[4,6,*]

1 Division of Biology, Kansas State University, Manhattan, KS 66506, USA
2 Allen County Community College, Burlingame, KS 66413, USA
3 Indian River State College, Fort Pierce, FL 34981, USA
4 Boyce Thompson Institute, Ithaca, NY 14853, USA
5 USDA-ARS, US Horticultural Research Laboratory, Fort Pierce, FL 34945, USA
6 Animal and Comparative Biomedical Sciences, University of Arizona, Tucson, AZ 85721, USA

## ABSTRACT

Chitinases are enzymes that digest the polysaccharide polymer chitin. During insect development, breakdown of chitin is an essential step in molting of the exoskeleton. Knockdown of chitinases required for molting is lethal to insects, making chitinase genes an interesting target for RNAi-based pest control methods. The Asian citrus psyllid, *Diaphorina citri*, carries the bacterium causing Huanglongbing, or citrus greening disease, a devastating citrus disease. We identified and annotated 12 chitinase family genes from *D. citri* as part of a community effort to create high-quality gene models to aid the design of interdictory molecules for pest control. We categorized the *D. citri* chitinases according to an established classification scheme and re-evaluated the classification of chitinases in other hemipterans. In addition to chitinases from known groups, we identified a novel class of chitinases present in *D. citri* and several related hemipterans that appears to be the result of horizontal gene transfer.

**Subjects** Genetics and Genomics, Animal Genetics, Bioinformatics

**Submitted:** 31 October 2021

\* Corresponding author. E-mail: suryasaha@cornell.edu

Preprint submitted at https://doi.org/10.1101/2021.10.30.466601

Included in the series: *Asian citrus psyllid community annotation* (https://doi.org/10.46471/GIGABYTE_SERIES_0001)

## DATA DESCRIPTION

During insect growth and development, the exoskeleton must be repeatedly shed and replaced. As part of this process, chitin, a polysaccharide polymer that is an important structural component of the cuticle, must be degraded [1]. Chitinases are enzymes that hydrolyze chitin into chitooligosaccharides, which can then be recycled to synthesize new chitin molecules [1, 2]. Restricting the degradation of chitin by inhibiting chitinases often results in lethality caused by molting defects (reviewed in [3]). Insect genomes usually contain 10–30 chitinase genes, with holometabolous insects generally having more than hemimetabolous insects [4]. These genes are often expressed in different stages and tissues, suggesting that they may have distinct roles during the life of the insect [2]. The various chitinase genes also encode proteins with different structures, particularly in the number of glycoside hydrolase 18 catalytic domains and chitin-binding domains (CBD). The most recent chitinase classification system, based on phylogenetic analysis and domain

conservation of proteins from 20 species, divides chitinases into 10 groups (I–X) [5]. Most of these groups appear to be ancient, with all but groups V and X being present in the ancestor of insects and crustaceans. This classification system has recently been applied to the chitinases of two hemipteran insects [6, 7]. These studies concluded that almost all the chitinase groups are represented in at least some hemipterans. However, group IX chitinases seem to have been lost from the hemipteran lineage. Several hemipteran chitinase genes that could not be definitively classified have been tentatively assigned to group IV.

## CONTEXT

We are part of a community that is manually curating genes from the genome of the Asian citrus psyllid, *Diaphorina citri* (Hemiptera: Liviidae; NCBI:txid121845), the vector of *Candidatus* Liberibacter asiaticus (*C*Las), the bacterium causing Huanglongbing (citrus greening disease) [8, 9]. The primary goal of this project is to create high-quality gene models of potential targets for gene-based pest control. The essential role of some chitinases during insect development makes them promising pest control targets. Several putative chitinase genes have previously been reported in *D. citri*, but these have not been manually curated [10]. Here, we report the annotation of the chitinase gene family in *D. citri*. We identified and annotated 11 chitinase genes, plus a gene encoding the related enzyme endo-beta-N-acetylglucosaminidase. We used phylogenetic and domain analyses to classify the chitinases according to the 10-group system established by Tetreau *et al.* [5]. Our results indicate that *D. citri* has a similar complement of chitinase genes to other hemipterans, but also has an unusual chitinase that seems to have arisen from a horizontal transfer event. Our phylogenetic analysis indicates that several hemipteran chitinases previously assigned to group IV are orthologous to this gene and should be reclassified.

## METHODS

*Diaphorina citri* chitinase genes were identified by BLAST analysis of *D. citri* sequences available on the Citrus Greening website [11] using orthologs from other insects as the query. To confirm orthology, we performed reciprocal BLASTs of the National Center for Biotechnology Information (NCBI) non-redundant protein database [12]. Genes were manually annotated in the *D. citri* v3 genome in Apollo (Apollo, RRID:SCR_001936; v2.1.0) using available evidence. A complete annotation workflow is available at protocols.io (Figure 1) [13].

Protein domains were identified using BLAST and InterPro (InterPro, RRID:SCR_006695) [14].

Phylogenetic analysis was performed with MEGA X (MEGA software, RRID:SCR_000667) [15]. Sequences were aligned with CLUSTALW (RRID:SCR_002909) [16] and trees were constructed by the neighbor-joining method with 1000 bootstrap replicates. Accession numbers for orthologs used in phylogenetic analysis are shown in Table 1. Counts for gene expression were obtained from the Citrus Greening Expression Network (CGEN) [17] and visualized using pheatmap (pheatmap, RRID:SCR_016418) [18] in R (R Project for Statistical Computing, RRID:SCR_001905) [19].

## DATA VALIDATION AND QUALITY CONTROL

We identified and annotated chitinase genes in the chromosome-level *D. citri* v3 genome (Table 2). BLAST analysis, domain content and phylogenetic analysis were used to

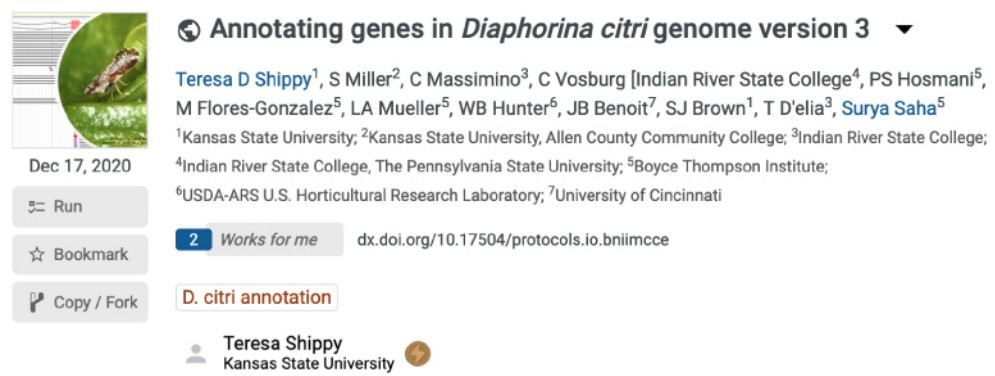

Annotating genes in *Diaphorina citri* genome version 3

Teresa D Shippy[1], S Miller[2], C Massimino[3], C Vosburg [Indian River State College[4], PS Hosmani[5], M Flores-Gonzalez[5], LA Mueller[5], WB Hunter[6], JB Benoit[7], SJ Brown[1], T D'elia[3], Surya Saha[5]

[1]Kansas State University; [2]Kansas State University, Allen County Community College; [3]Indian River State College; [4]Indian River State College, The Pennsylvania State University; [5]Boyce Thompson Institute; [6]USDA-ARS U.S. Horticultural Research Laboratory; [7]University of Cincinnati

Dec 17, 2020

**Figure 1.** Protocols.io protocol for annotating genes in *Diaphorina citri* genome version 3. https://www.protocols.io/widgets/doi?uri=dx.doi.org/10.17504/protocols.io.bniimcce

determine the orthology of annotated genes. We followed the established convention for naming chitinase genes, using the same name as the *Drosophila melanogaster* ortholog whenever possible [20].

## Group I chitinases

Group I chitinases contain one catalytic domain and one C-terminal CBD (Figure 2) [2]. Most insects have a single group I chitinase (Table 3), which is typically named Chitinase 5 (Cht5) (Table 4). However, multiple group I chitinase genes have been found in mosquitoes [21], as well as in several hemimetabolous insects [4, 7, 22, 23]. Within the Hemiptera, *Acyrthosiphon pisum* and *Bemisia tabaci* have one *Cht5* ortholog, while *Nilaparvata lugens* and *Sogatella furcifera* have two [4, 6, 7, 23]. We identified only one *Cht5* gene in the *D. citri* genome (Tables 2 and 3, Figure 3). As expected, it encodes a protein with one catalytic domain and one CBD.

Chitinase groups are based on the classification system established by Tetreau *et al.* [5], except for ChtPE, which is described in this work. *D. citri* gene numbers were determined based on our annotation of the *D. citri* v3 genome. Counts in other insects are based on the literature [4, 6, 7, 21, 23] and our phylogenetic analysis.

## Group II chitinases

Group II chitinases are typically named Chitinase 10 (Cht10) in insects (Table 4) [2]. These chitinases are high-molecular-weight chitinases with multiple catalytic domains (some active and some inactive) and several CBDs [2]. Most previously studied insects have only one *Cht10* gene (Table 3), although two were found in *N. lugens* (*NlCht10* and *NlCht1*) [23]. Two of the chitinases we annotated in *D. citri* cluster with the Cht10 proteins during phylogenetic analysis. One of these, Cht10-1, is a typical Cht10 protein. It is a large, 21-exon gene that encodes a protein containing five catalytic domains and two CBDs. The second protein identified as a potential Cht10 in *D. citri* is much smaller and only contains a catalytic domain. Despite the difference in size and domain content, phylogenetic analysis indicates this protein is most closely related to the Cht10 proteins, so we have named it Cht10-2 (Figure 3). Interestingly, the *B. tabaci* Cht4 protein, which had been tentatively placed in group IV [6], also has only a catalytic domain and clusters with the group II chitinases in our tree. Thus, we suggest that this should be reassigned to group II

**Table 1.** Accession numbers of proteins used in phylogenetic analysis.

| Name in tree | Order | Species | Accession |
|---|---|---|---|
| TcCht1 | Coleoptera | *Tribolium castaneum* | XP_971647.1 |
| TcCht2 | Coleoptera | *Tribolium castaneum* | XP_970191.2 |
| TcCht3 | Coleoptera | *Tribolium castaneum* | EFA08056.1 |
| TcCht4 | Coleoptera | *Tribolium castaneum* | NP_001073567.1 |
| TcCht5 | Coleoptera | *Tribolium castaneum* | NP_001034524.1 |
| TcCht6 | Coleoptera | *Tribolium castaneum* | XP_967813.1 |
| TcCht7 | Coleoptera | *Tribolium castaneum* | NP_001036035.1 |
| TcCht8 | Coleoptera | *Tribolium castaneum* | NP_001038094.1 |
| TcCht9 | Coleoptera | *Tribolium castaneum* | NP_001038096.1 |
| TcCht10 | Coleoptera | *Tribolium castaneum* | NP_001036067.1 |
| TcCht11 | Coleoptera | *Tribolium castaneum* | NP_001038095.1 |
| TcCht12 | Coleoptera | *Tribolium castaneum* | XP_972802.2 |
| TcCht13 | Coleoptera | *Tribolium castaneum* | NP_001036034.1 |
| TcCht14 | Coleoptera | *Tribolium castaneum* | XP_973005.1 |
| TcCht15 | Coleoptera | *Tribolium castaneum* | XP_973077.1 |
| TcCht16 | Coleoptera | *Tribolium castaneum* | NP_001034515.1 |
| TcCht17 | Coleoptera | *Tribolium castaneum* | XP_972719.1 |
| TcCht18 | Coleoptera | *Tribolium castaneum* | XP_973161.2 |
| TcCht19 | Coleoptera | *Tribolium castaneum* | XP_973119.2 |
| TcCht20 | Coleoptera | *Tribolium castaneum* | NP_001034516.3 |
| TcCht21 | Coleoptera | *Tribolium castaneum* | NP_001034517.1 |
| TcIDGF2 | Coleoptera | *Tribolium castaneum* | NP_001038092.1 |
| TcIDGF4 | Coleoptera | *Tribolium castaneum* | NP_001038091.1 |
| TcENGase | Coleoptera | *Tribolium castaneum* | EFA09314.2 |
| DmCht1 | Diptera | *Drosophila melanogaster* | NP_609190.2 |
| DmCht10 | Diptera | *Drosophila melanogaster* | EAA46011.1 |
| DmCht11 | Diptera | *Drosophila melanogaster* | NP_572361.1 |
| DmCht12 | Diptera | *Drosophila melanogaster* | NP_726022.1 |
| DmCht2 | Diptera | *Drosophila melanogaster* | NP_477298.2 |
| DmCht4 | Diptera | *Drosophila melanogaster* | NP_524962.2 |
| DmCht5 | Diptera | *Drosophila melanogaster* | NP_650314.1 |
| DmCht6 | Diptera | *Drosophila melanogaster* | NP_572598.3 |
| DmCht7 | Diptera | *Drosophila melanogaster* | NP_647768.3 |
| DmCht8 | Diptera | *Drosophila melanogaster* | NP_611542.2 |
| DmCht9 | Diptera | *Drosophila melanogaster* | NP_611543.3 |
| DmIDGF1 | Diptera | *Drosophila melanogaster* | NP_477258.1 |
| DmIDGF2 | Diptera | *Drosophila melanogaster* | NP_477257.2 |
| DmIDGF3 | Diptera | *Drosophila melanogaster* | NP_723967.1 |
| DmIDGF4 | Diptera | *Drosophila melanogaster* | NP_727374.1 |
| DmIDGF5 | Diptera | *Drosophila melanogaster* | NP_611321.3 |
| DmIDGF6 | Diptera | *Drosophila melanogaster* | NP_477081.1 |
| DcCht5 | Hemiptera | *Diaphorina* citri | Dcitr06g10380.1.1 |
| DcCht7 | Hemiptera | *Diaphorina* citri | Dcitr07g07740.1.1 |
| DcIDGF1 | Hemiptera | *Diaphorina* citri | Dcitr02g06220.1.1 |
| DcIDGF2 | Hemiptera | *Diaphorina* citri | Dcitr02g06220.1.1 |
| DcIDGF3 | Hemiptera | *Diaphorina* citri | Dcitr02g06590.1.1 |
| DcCht6 | Hemiptera | *Diaphorina* citri | Dcitr10g04150.1.1 |
| DcCht11 | Hemiptera | *Diaphorina* citri | Dcitr01g03820.1.1 |
| DcCht3 | Hemiptera | *Diaphorina* citri | Dcitr07g08380.1.1 |
| DcENGase | Hemiptera | *Diaphorina* citri | Dcitr01g14510.1.1 |
| DcChtPE | Hemiptera | *Diaphorina* citri | Dcitr11g03190.1.1 |
| DcCht10-1 | Hemiptera | *Diaphorina* citri | Dcitr02g11110.1.1 |
| DcCht10-2 | Hemiptera | *Diaphorina* citri | Dcitr12g04430.1.1 |
| ApCht1 | Hemiptera | *Acyrthosiphon pisum* | NP_001162142.1 |
| ApCht2 | Hemiptera | *Acyrthosiphon pisum* | XP_001943038.2 |

**Table 1.** (Continued)

| Name in tree | Order | Species | Accession |
|---|---|---|---|
| ApCht3 | Hemiptera | *Acyrthosiphon pisum* | XP_029343203.1 |
| ApCht4 | Hemiptera | *Acyrthosiphon pisum* | XP_001950380.1 |
| ApCht5 | Hemiptera | *Acyrthosiphon pisum* | XP_008181779.1 |
| ApCht6 | Hemiptera | *Acyrthosiphon pisum* | XP_008182857.1 |
| ApCht7 | Hemiptera | *Acyrthosiphon pisum* | XP_008183766.1 |
| ApCht8 | Hemiptera | *Acyrthosiphon pisum* | XP_001945470.2 |
| ApENGase | Hemiptera | *Acyrthosiphon pisum* | XP_016658011.1 |
| NlCht1(partial) | Hemiptera | *Nilaparvata lugens* | AJO25036.1 |
| NlCht2 | Hemiptera | *Nilaparvata lugens* | AJO25037.1 |
| NlCht3 | Hemiptera | *Nilaparvata lugens* | AJO25038.1 |
| NlCht4 | Hemiptera | *Nilaparvata lugens* | AJO25039.1 |
| NlCht5 | Hemiptera | *Nilaparvata lugens* | AJO25040.1 |
| NlCht6 | Hemiptera | *Nilaparvata lugens* | AJO25041.1 |
| NlCht7 | Hemiptera | *Nilaparvata lugens* | AJO25042.1 |
| NlCht8 | Hemiptera | *Nilaparvata lugens* | AJO25043.1 |
| NlCht10 | Hemiptera | *Nilaparvata lugens* | AJO25045.1 |
| NlIDGF | Hemiptera | *Nilaparvata lugens* | AJO25056.1 |
| NlENGase | Hemiptera | *Nilaparvata lugens* | AJO25057.1 |
| BtCht2 | Hemiptera | *Bemisia tabaci* | UDL18255.1 |
| BtCht3 | Hemiptera | *Bemisia tabaci* | UDL18256.1 |
| BtCht4 | Hemiptera | *Bemisia tabaci* | UDL18257.1 |
| BtCht5 | Hemiptera | *Bemisia tabaci* | UDL18258.1 |
| BtCht6 | Hemiptera | *Bemisia tabaci* | UDL18259.1 |
| BtCht7 | Hemiptera | *Bemisia tabaci* | UDL18260.1 |
| BtCht8 | Hemiptera | *Bemisia tabaci* | UDL18261.1 |
| BtCht9 | Hemiptera | *Bemisia tabaci* | UDL18262.1 |
| BtCht10 | Hemiptera | *Bemisia tabaci* | UDL18263.1 |
| BtCht11 | Hemiptera | *Bemisia tabaci* | XP_018912124.1 |
| BtIDGF1 | Hemiptera | *Bemisia tabaci* | UDL18264.1 |
| BtIDGF2 | Hemiptera | *Bemisia tabaci* | UDL18265.1 |
| BtIDGF3 | Hemiptera | *Bemisia tabaci* | UDL18266.1 |
| BtENGase | Hemiptera | *Bemisia tabaci* | UDL18267.1 |
| TuXP015788124.1 | Trombidiformes | *Tetranychus urticae* | XP_015788124.1 |
| SfXP025409901.1 | Hemiptera | *Sipha flava* | XP_025409901.1 |
| DnXP015372246.1 | Hemiptera | *Diuraphis noxia* | XP_015372246.1 |
| MpXP022167894.1 | Hemiptera | *Myzus persicae* | XP_022167894.1 |
| ArCAF1372083.1 | Bdelloida | *Adineta ricciae* | CAF1372083.1 |
| BcXP037026665.1 | Diptera | *Bradysia coprophila* | XP_037026665.1 |
| CnXP031616960.1 | Diptera | *Contarinia nasturtii* | XP_031616960.1 |
| AcCht-h | Lepidoptera | *Agrius convolvuli* | BAE16588.1 |
| BmCht-h | Lepidoptera | *Bombyx mori* | XP_037867787.1 |
| DpCht-h | Lepidoptera | *Danaus plexippus plexippus* | XP_032522474.1 |
| PxCht-h | Lepidoptera | *Papilio xuthus* | KPJ01281.1 |
| SlCht-h | Lepidoptera | *Spodoptera litura* | XP_022815620.1 |
| OfCht-h | Lepidoptera | *Ostrinia furnacalis* | XP_028158980.1 |

Ortholog names used in the phylogenetic tree (Figure 3), taxonomic order, species name and accession number are shown.

(Tables 3 and 4). NlCht10, one of the *N. lugens* proteins classified as a group II chitinase [23], surprisingly clusters with the *Drosophila* and *Tribolium* group VI proteins in our tree (Figure 3). The high level of sequence identity between NlCht10 and NlCht1, however, indicates that NlCht10 should remain in group II. These conflicting phylogenetic results suggest that additional analysis of the *N. lugens* group II chitinases is warranted.

**Table 2.** Manually annotated chitinase family genes from *Diaphorina citri*.

| Group | Gene/Isoform | OGSv3 ID | Evidence supporting annotation | | | |
|---|---|---|---|---|---|---|
| | | | MCOT | Iso-Seq | RNA-Seq | Ortholog |
| I | *Chitinase 5* | Dcitr06g10380.1.1 | MCOT12176.1.CO | X | X | X |
| II | *Chitinase 10-1* | Dcitr02g11110.1.1 | MCOT12469.0.CO | | X | X |
| II | *Chitinase 10-2* | Dcitr12g04430.1.1 | MCOT05985.1.CT | | | X |
| III | *Chitinase 7* | Dcitr07g07740.1.1 | MCOT01854.1.CT | X | X | X |
| V | *Imaginal disc growth factor 1* | Dcitr02g06220.1.1 | | X | X | X |
| V | *Imaginal disc growth factor 2* | Dcitr02g06210.1.1 | MCOT17201.0.CT | | X | X |
| V | *Imaginal disc growth factor 3* | Dcitr02g06590.1.1 | | X | X | X |
| VI | *Chitinase 6* | Dcitr10g04150.1.1 | MCOT02473.0.CO | X | | X |
| | | Dcitr10g04150.1.2 | | | | |
| VIII | *Chitinase 11* | Dcitr01g03820.1.1 | | X | X | X |
| X | *Chitinase 3* | Dcitr07g08380.1.1 | MCOT14388.2.CO | | X | X |
| ENGase | *endo-beta-N-acetylglucosaminidase* | Dcitr01g14510.1.1 | MCOT20578.0.CT | | X | X |
| ChtPE | | Dcitr11g03190.1.1 | MCOT00573.0.CT | | X | X |

The chitinase group, OGSv3 gene identifier and evidence types used during the annotation process are listed for each gene. MCOT identification numbers denote models from the Maker, Cufflinks, Oases and Trinity transcriptome [8].

**Table 3.** Estimated number of chitinase genes in various insect species.

| Species | Chitinase groups | | | | | | | | | | | | |
|---|---|---|---|---|---|---|---|---|---|---|---|---|---|
| | I | II | III | IV | V | VI | VII | VIII | IX | X | ENGase | ChtPE | Total |
| *D. melanogaster* | 1 | 1 | 1 | 4 | 6 | 1 | 1 | 1 | 1 | 0 | 1 | 0 | 18 |
| *A. gambiae* | 5 | 1 | 1 | 8 | 2 | 1 | 1 | 1 | 1 | 0 | 1 | 0 | 22 |
| *T. castaneum* | 1 | 1 | 1 | 14 | 2 | 1 | 1 | 1 | 1 | 1 | 1 | 0 | 25 |
| *S. furcifera* | 2 | 1 | 1 | 0 | 2 | 1 | 1 | 1 | 0 | 1 | 1 | 0 | 11 |
| *N. lugens* | 2 | 2 | 1 | 0 | 2 | 1 | 1 | 1 | 0 | 1 | 1 | 0 | 12 |
| *B. tabaci* | 1 | 2 | 1 | 0 | 3 | 1 | 1 | 1 | 0 | 1 | 1 | 2 | 14 |
| *A. pisum* | 1 | 1 | 1 | 0 | 1 | 1 | 0 | 1 | 0 | 1 | 1 | 1 | 9 |
| *D. citri* | 1 | 2 | 1 | 0 | 3 | 1 | 0 | 1 | 0 | 1 | 1 | 1 | 12 |

## Group III chitinases

The group III chitinases are typically named Chitinase 7 (Cht7) in insects (Table 4) [2]. Most insects have one Cht7 that contains an N-terminal transmembrane domain, plus two catalytic domains followed by a CBD (Figure 2) [20]. In *D. citri*, we identified one *Cht7* gene (Tables 2 and 3). As expected, the predicted protein contained two catalytic domains, followed by one CBD (Figure 2). Like the *A. pisum* and *S. furcifera* group III chitinases, DcCht7 has an N-terminal signal peptide [4, 7], suggesting that at least some hemipteran group III chitinases may be secreted and thus function differently than their orthologs in holometabolous insects that have an N-terminal transmembrane domain.

## Group IV chitinases

In holometabolous insects, group IV is the largest and most diverse group of chitinases [2]. These chitinases have the greatest variation in domain organization and are found in clusters in some insect genomes, suggesting duplication events. In hemimetabolous insects, group IV has previously been used as a catch-all group for chitinases that could not be clearly assigned to a group [6, 23]. However, recently, several of the hemipteran chitinases previously assigned to group IV have been reclassified as group X chitinases [6]. Moreover,

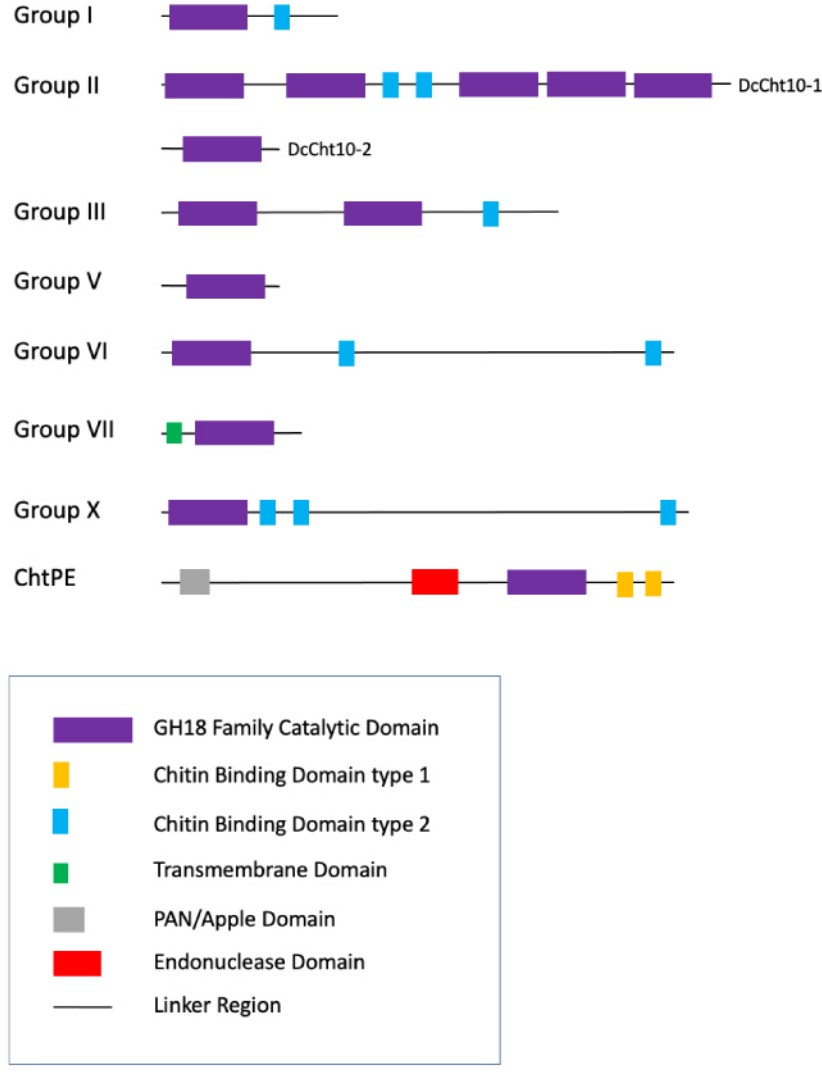

**Figure 2.** Chitinase domain organization in *Diaphorina citri*. Chitinases are categorized by group based on phylogenetic analysis, sequence similarity, and domain organization. *D. citri* domain analysis was performed with InterPro. The two Group II proteins with different domain structures are both shown. Group V represents three proteins with the same domain structure: Idgf1, Idgf2 and Idgf3.

in our phylogenetic analysis (Figure 3), no *D. citri* chitinases clustered with group IV, and the other hemipteran chitinases that had previously been placed in group IV (*B. tabaci* Cht8 and Cht9) were part of a novel cluster discussed in more detail below. These observations suggest that hemipterans lack group IV chitinases.

## Group V chitinases

The group V chitinases were first identified for their role in the growth of imaginal disc tissue in *Drosophila* and were named Imaginal disc growth factors (Idgf) [2, 25]. *D. melanogaster* has six *Idgf* genes, but most insects have fewer (Tables 3 and 4). Phylogenetic analysis suggests that there have been several independent duplications of *Idgf* genes in insect lineages [4]. In *D. citri*, we identified three *Idgf* genes (Tables 2 and 3),

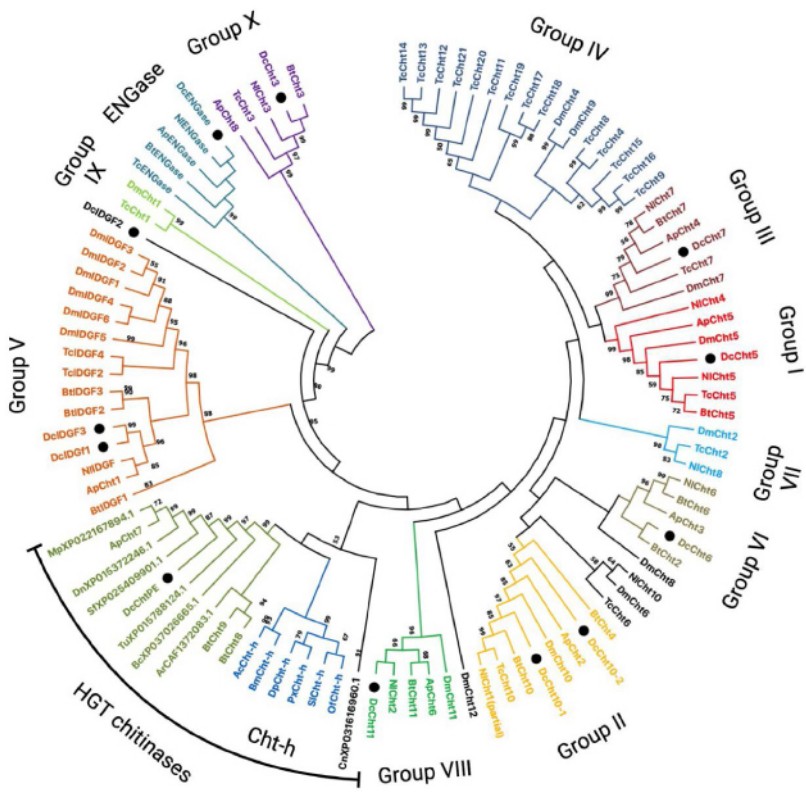

**Figure 3.** Phylogenetic tree of chitinase and chitinase-like family members. CLUSTALW was used to perform multiple sequence alignments. The tree was constructed with MEGA X software using neighbor-joining analysis (1000 bootstrap replicates). Bootstrap values below 50 are not shown. The final annotated tree graphic was created with BioRender.com [24]. Proteins used in tree construction are from Diptera: *Drosophila melanogaster* (Dm), *Anopheles gambiae* (Ag), *Bradysia coprophila* (Bc), *Contarinia nasturtii* (Cn); Lepidoptera: *Agrius convolvuli* (Ac), *Bombyx mori* (Bm), *Danaus plexippus plexippus* (Dp), *Papilio xuthus* (Px), *Spodoptera litura* (Sl), *Ostrinia furnacalis* (Of); Coleoptera: *Tribolium castnaeum* (Tc); Hemiptera: *Nilaparvata lugens* (Nl), *Acyrthosiphon pisum* (Ap), *Bemisia tabaci* (Bt), *Diaphorina citri* (Dc), *Sipha flava* (Sf), *Myzus persicae* (Mp), *Diuraphis noxia* (Dn), Arachnida: *Tetranychus urticae* (Tu); Rotifera: *Adineta ricciae* (Ar). *D. citri* proteins from genes annotated in this work are marked with black circles. Colors delineate chitinase groups which are also labeled. Genes that do not cluster well with any group are black.

which we have named *Idgf1*, *Idgf2* and *Idgf3*. These genes are not one-to-one orthologs of the *Drosophila Idgf1*, *Idgf2* and *Idgf3* genes, as phylogenetic analysis suggests that *Idgf* genes have duplicated independently in these two insect lineages (Figure 3). All three *Idgf* genes in *D. citri* are found in a 1.25-megabase pair (Mbp) region of chromosome 2, with *Idgf1* and *Idgf2* adjacent to one another on the same strand. Idgf1 and Idgf3 form their own clade in our phylogenetic tree, while Idgf2 is an outgroup to the other group V chitinases, suggesting it has diverged more extensively than the other two paralogs (Figure 3).

As seen in group V chitinases of other insects, all three *D. citri* Idgf proteins have only one catalytic domain and they do not contain a CBD (Figure 2). The catalytic domain of Idgf proteins is inactive because of a mutation that produces an aspartic acid to alanine substitution in conserved motif II [2, 26]. This mutation is present in all three *D. citri Idgf* genes, confirming their identity.

**Table 4.** Insect chitinase orthologs.

| | Dm | Ag | Ms | Tc | Sf | Nl | Bt | Ap | Dc |
|---|---|---|---|---|---|---|---|---|---|
| **Group 1** | Cht5 | Cht5-1 | Cht5 | Cht5 | Cht5 | Cht5 | Cht5 | Cht5 | Cht5 |
| | | Cht5-2 | | | | | | | |
| | | Cht5-3 | | | | | | | |
| | | Cht5-4 | | | | | | | |
| | | Cht5-5 | | | | | | | |
| | | | | | Cht4 | Cht4 | | | |
| **Group 2** | Cht10 | Cht10 | Cht10 | Cht10 | Cht10 | Cht10 | | Cht2 | Cht10-1 |
| | | | | | | Cht1 | Cht10 | | |
| | | | | | | | *Cht4* | | |
| | | | | | | | | | Cht10-2 |
| **Group 3** | Cht7 | Cht7 | Cht7 | Cht7 | Cht7 | Cht7 | Cht7 | Cht4 | Cht7 |
| **Group 4** | Cht4 | Cht4 | Cht8 | Cht4 | | | | | |
| | Cht8 | Cht8 | | Cht8 | | | | | |
| | Cht9 | Cht9 | | Cht9 | | | | | |
| | Cht12 | Cht12 | | Cht12 | | | | | |
| | | Cht13 | | Cht13 | | | | | |
| | | | | Cht14 | | | | | |
| | | | | Cht15 | | | | | |
| | | Cht16 | | Cht16 | | | | | |
| | | | | Cht17 | | | | | |
| | | | | Cht18 | | | | | |
| | | | | Cht19 | | | | | |
| | | | | Cht20 | | | | | |
| | | | | Cht21 | | | | | |
| | | | | Cht22 | | | | | |
| | | Cht23 | | | | | | | |
| | | Cht24 | | | | | | | |
| **Group 5** | IDGF1 | | | | | | | | |
| | IDGF2 | | | | | | | | |
| | IDGF3 | | | | | | | | |
| | IDGF4 | | IDGF1 | | IDGF1 | | IDGF1 | | |
| | IDGF5 | | | | IDGF2 | | IDGF2 | | |
| | IDGF6 | | | | | | IDGF3 | | |
| | | IDGF2 | | | | | | | |
| | | IDGF4 | | | | | | | |
| | | | | IDGF2 | | | | | |
| | | | | IDGF4 | | | | | |
| | | | | | | IDGF | | | |
| | | | | | | Cht9 | | | |
| | | | | | | | | Cht1 | |
| | | | | | | | | | IDGF1 |
| | | | | | | | | | IDGF2 |
| | | | | | | | | | IDGF3 |
| **Group 6** | Cht6 | Cht6 | Cht6 | Cht6 | Cht6 | Cht6 | Cht6 | Cht3 | Cht6 |
| | | | | | | | *Cht2?* | | |
| **Group 7** | Cht2 | Cht2 | Cht2 | Cht2 | Cht8 | Cht8 | | | |
| **Group 8** | Cht11 | Cht11 | Cht11 | Cht11 | Cht2 | Cht2 | Cht11 | Cht6 | Cht11 |
| **Group 9** | DmCht1 | | Cht1 | Cht11 | | | | | |
| **Group 10** | | | Cht3 | Cht3 | Cht3 | Cht3 | Cht3 | Cht8 | Cht3 |
| **ENGase** | CG5613 | XP 310876.4 | | XP 008197368.1 | EnGase | ENGase | ENGase | ENGase | ENGase |

| Table 4. (Continued) | | | | | | | | | |
|---|---|---|---|---|---|---|---|---|---|
| | **Dm** | **Ag** | **Ms** | **Tc** | **Sf** | **Nl** | **Bt** | **Ap** | **Dc** |
| **Sl-Clp** | CG8460 | XP 317335.2 | | XP 971647.1 | | | | | |
| **ChtPE** | | | | | | | | *Cht7* | PE |
| | | | | | | | *Cht8* | | |
| | | | | | | | *Cht9* | | |

Revised group assignment of chitinase proteins from *Drosophila melanogaster* (Dm), *Anopheles gambiae* (Ag), *Manduca sexta* (Ms), *Tribolium castaneum* (Tc), *Sogatella furcifera* (Sf), *Nilaparvata lugens* (Nl), *Bemisia tabaci* (Bt), *Acyrthosiphon pisum* (Ap) and *Diaphorina citri* (Dc) based the analysis described in this work. A blank cell means no members of a particular group have been identified in that insect. Orthologs shown in italics indicate changes in group assignment based on our analysis. A question mark denotes uncertainty in the new classification.

## Group VI Chitinases

In insects, the group VI chitinases are usually named Chitinase 6 (Cht6) (Table 4) [2]. In holometabolous insects, group VI chitinases have a similar domain structure to group I chitanases with an N-terminal catalytic domain and one CBD, but additionally have a long serine/threonine (S/T)-rich region at the C-terminus [2]. The hemipterans *N. lugens* and *A. pisum* each have a single group VI chitinase. These proteins differ from their holometabolous orthologs in that they have a second CBD near the C-terminus [4, 23]. In *D. citri*, we identified one *Cht6* gene that also encodes a protein with a second CBD (Figure 2). The *D. citri* Cht6 protein also contains a long stretch of amino acids between the CBDs, which contains approximately 25% S/T residues, supporting its classification as a group VI chitinase. We identified two isoforms of Cht6 in *D. citri*, which differ only in the length of the S/T-rich region between the CBDs. Similar isoforms have been reported for *S. furcifera* Cht6 [7].

In contrast to the other chitinase groups, the group VI orthologs do not all cluster together in our phylogenetic tree (Figure 3). The hemipteran group VI proteins form one cluster, while the *T. castaneum* and *D. melanogaster* Cht6 orthologs are in a separate cluster with *N. lugens* Cht10, which has been classified in group II [23]. BtCht2, which was formerly classified as group VII [6], also clusters with the group VI genes, albeit with low bootstrap values (Figure 3). Moreover, *D. melanogaster* Cht8, which is considered a group IV member, is the closest outgroup to the hemipteran group VI proteins.

## Group VII chitinases

Group VII chitinases are typically named Chitinase 2 (Cht2) in insects [2]. Within hemipterans, the planthoppers *N. lugens* and *S. furcifera* have a group VII chitinase gene [7, 23], but *A. pisum* does not (Table 3) [4]. *B. tabaci* was reported to have a group VII gene, which was consequently named *BtCht2* [6]. However, the placement of *BtCht2* in group VII was only weakly supported by phylogenetic analysis and, in our phylogenetic tree (Figure 3), it clusters with the group VI genes as discussed above. Although the proper classification of *BtCht2* is unclear, our interpretation is that *B. tabaci* lacks a group VII gene (Table 4). Likewise, we found no group VII chitinase gene in the genome of *D. citri*. Since the three hemipterans lacking group VII genes are all sternorrhyncans, these results suggest that the group VII chitinase may have been lost after the divergence of the Sternorrhynca from other hemipterans.



### Group VIII chitinases

Group VIII chitinases are typically called Chitinase 11 (Cht11) in insects (Table 4) [2]. To our knowledge, all insects examined to date have only one group VIII chitinase gene. We too identified only one group VIII chitinase in the *D. citri* genome. Like several other group VIII chitinases, *D. citri* Cht11 has an N-terminal transmembrane domain and a catalytic domain, but no CBD [2, 4].

### Group IX chitinases

Group IX chitinases appear to be an ancient group, since orthologs are found in organisms as distantly related to arthropods as sea urchins and nematodes [5]. However, no group IX chitinases have been found in hemipteran genomes thus far [4, 6, 7, 23]. As expected, we were also unable to identify a group IX gene in *D. citri* (Tables 3 and 4).

### Group X chitinases

Group X chitinases, most of which are named Cht3 (Table 4), were first recognized as a separate group by Tetreau *et al.* [5]. Several members of this new group had previously been assigned to group IV, although their membership in that group was always uncertain. Group X genes are found only in arthropods and seem to have been lost in the dipteran lineage [5]. The proteins encoded by group X genes have a unique, highly conserved structure consisting of a single catalytic domain followed by two closely spaced CBDs, a long intervening region with many potential glycosylation sites, and a third CBD near the C-terminus [5–7, 23]. We identified and annotated one *Cht3* gene in *D. citri*. The encoded protein clusters with group X members in our phylogenetic analysis (Figure 3) and shares the same domain structure (Figure 2).

### ENGases

The endo-beta-N-acetylglucosaminidase (ENGase) proteins are part of the GH18 chitinase-like superfamily, and have therefore been included in recent phylogenetic analyses of chitinases [4, 23]. Like the group V chitinases, these proteins lack chitinase activity because of a change in the catalytic domain. *ENGase* orthologs have been found in various insects, including in hemipterans [4, 6, 7, 23]. In the *D. citri* genome, we identified one *ENGase* ortholog (Tables 2, 3 and 4).

### Chitinase PE

*D. citri* has one chitinase gene that could not be classified based on the currently defined groups. In our tree, it clusters with *A. pisum* Cht7, which also has not been definitively classified [4], and *B. tabaci* Cht8 and Cht9, which had been tentatively included in group IV [6].

The *A. pisum* and *D. citri* proteins have an unusual structure: an N-terminal signal peptide, a long N-terminal region where the only recognizable sequence is a PAN/Apple domain, and a DNA/RNA non-specific endonuclease domain in the central portion of the protein, followed by the chitinase catalytic domain and multiple CBDs. We have named the *D. citri* gene *Chitinase PE* (*ChtPE*) to denote the presence of the P̲AN domain and e̲ndonuclease domain.

Previously, it was noted that the three CBDs in *A. pisum* Cht7 are ChtBD1-type domains (typically found in plants and fungi) rather than the ChtBD2 type found in other insect



chitinases [4]. We analyzed the domain structure of *D. citri* ChtPE and *B. tabaci* Cht8 and Cht9 and found that these proteins also have ChtBD1 domains, although the *D. citri* protein has only two.

BLAST analysis suggests that these novel chitinases have a very unusual phylogenetic distribution. Within the Hemiptera, they are present in several, but not all, of the sequenced genomes from sternorrhyncans (aphids, psyllids and whiteflies). Orthologous genes encoding all the domains found in ChtPE are also found in a few other phylogenetically dispersed insects, as well as in several spider mites, springtails and rotifers.

The presence of plant/fungi-like CBDs and the limited phylogenetic distribution of the gene suggest that *ChtPE* may have arisen by horizontal gene transfer (HGT), although the source of the gene is not clear. There have been previous reports of HGT involving chitinases. Many lepidopterans have a *Cht-h* gene that seems to have been horizontally transferred from bacteria [5]. A separate instance of HGT of a bacterial chitinase has been reported in spider mites [27]. However, BLAST analysis, domain content and phylogenetic analysis show that these proteins are clearly distinct from ChtPE (Figure 3).

It is unclear how the phylogenetic distribution of *ChtPE*-like genes arose, since this would seem to require either horizontal transfer into multiple lineages, or an ancient horizontal transfer followed by loss in most lineages. Neither scenario is particularly parsimonious. The presence of *ChtPE*-like genes in several sternorrhynchans but very few other hemipterans suggests there may have been a horizontal transfer event early in the sternorrhyncan lineage. However, it is unclear whether the *B. tabaci* genes *BtCht8* and *BtCht9* are orthologous to *ChtPE*. *BtCht8* and *BtCht9* are unusual in that they are single exon genes [6], while the related *A. pisum* and *D. citri* genes have multiple exons. Moreover, the encoded *B. tabaci* proteins have the chitinase catalytic domain and the ChtBD1 domains, but lack the PAN/Apple and endonuclease domains. Regardless of the number of HGT events, *A. pisum Cht7*, *BtCht8* and *BtCht9* belong with the HGT chitinases (Table 4) rather than in group IV where the *B. tabaci* proteins were previously placed [6].

## Expression of chitinase genes in *D. citri*

We assessed expression of the chitinase genes in *D. citri* using the Citrus Greening Expression Network [17] found on the Citrus Greening website [11] (Figure 4, Table 5). This tool allows comparison of gene expression levels in various publicly available *D. citri* RNA-seq datasets that vary by life stage, tissue, food source, and *C*Las exposure. In *D. citri*, *Cht5*, *Cht10-1*, and *Cht11* are expressed at highest levels in eggs with somewhat lower levels in nymphs, while *Cht3*, *Cht6*, and *Cht7* are most highly expressed in nymphs. The unusual group II gene *Cht10-2* is expressed at low-to-moderate levels in all stages and in most tissues. *IDGF2* expression is mostly restricted to eggs, while *IDGF1* and *IDGF3* are expressed at all stages, but highest in adults. *ENGase* shows low levels of expression in all samples, with the highest expression in eggs and female abdomens. A few of the chitinases (*Cht5*, *Cht11*, *IDGF1* and *IDGF3*) show moderate expression in the gut. ChtPE is expressed in all stages and tissues, with the highest expression in head, thorax and midgut. These expression trends are consistent with reports from other hemipterans, particularly for the stage showing the highest expression for each gene [4, 6, 7, 23].

That chitinase genes in hemipterans are generally conserved suggests that the genes may also have conserved functions. Based on expression data and RNAi studies in other insects, including several hemipterans [2, 6, 7, 23], the *D. citri Cht5*, *Cht7 and Cht10* orthologs are the

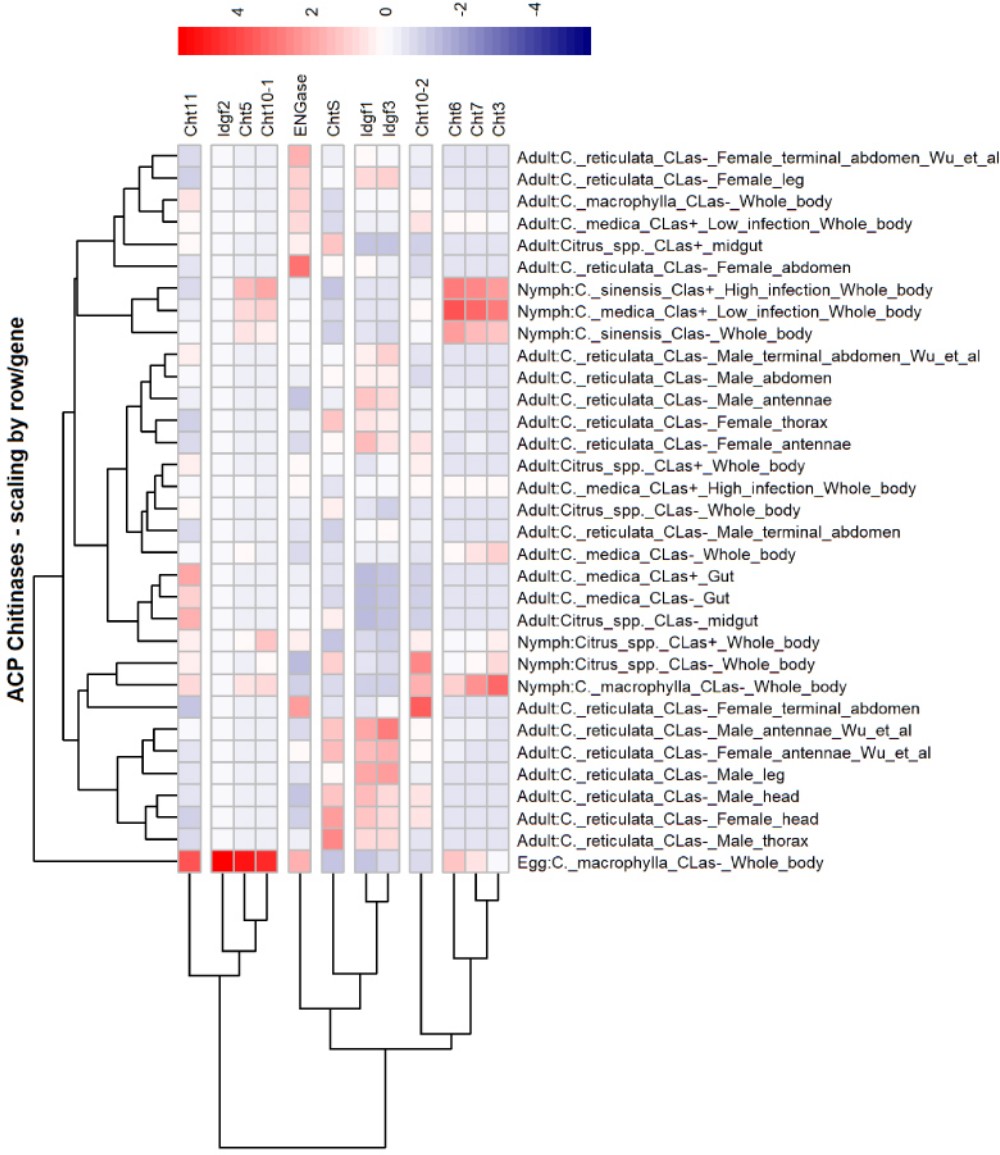

**Figure 4.** Expression of chitinase genes in *Diaphorina citri*. The heatmap was created from publicly available RNA-Seq expression data ([28–32] and NCBI Bioprojects PRJNA609978 and PRJNA448935) obtained from CGEN [17]. Expression is scaled by gene. Hierarchical clustering has been applied to both genes and RNA-seq samples such that those with similar expression are grouped together. Expression data used to create the heat map are provided in Table 4.

most likely to be required for molting during development. Thus, these genes should be prioritized as potential targets for RNAi-based pest control. Knockdown of the other chitinase genes will probably have only subtle effects, possibly because of redundancy, and understanding the function of these genes will require more extensive analysis. While this manuscript was under review, Wu *et al.* [33] published an independent characterization of *D. citri* chitinase genes with very similar results. They performed RNAi with each of the genes and, as we predicted, found that only *DcCht5*, *DcCht7*, *DcCht10-1* and *DcCht10-2* affected molting.

**Table 5.** Expression counts of *Diaphorina citri* chitinase genes.

| Gene ID | Cht5 | Cht10-1 | Cht10-2 | Cht7 | ChtPE | Idgf1 | Idgf2 | Idgf3 | Cht6 | Cht11 | Cht3 | ENGase |
|---|---|---|---|---|---|---|---|---|---|---|---|---|
| Egg: *C. macrophylla* CLas– Whole body | 80.44 | 17.67 | 5.15 | 181.38 | 6.15 | 185.72 | 110.96 | 461.11 | 22.37 | 108.89 | 17.17 | 15.86 |
| Nymph: *C. medica* CLas+ Low infection Whole body | 17 | 5.07 | 22.3 | 598.95 | 30.46 | 580.75 | 1.47 | 615.06 | 53.69 | 31.59 | 125.87 | 7.96 |
| Nymph: *C. sinensis* CLas+ High infection Whole body | 26.56 | 8.01 | 15.54 | 505.99 | 16.6 | 594.6 | 2.09 | 611.67 | 42.55 | 25.11 | 98.39 | 6.42 |
| Nymph: *C. sinensis* CLas– Whole body | 14.85 | 2.45 | 17.26 | 305.92 | 27.69 | 615.89 | 0.58 | 423.62 | 34.1 | 38.01 | 68.42 | 8.25 |
| Nymph: *C. macrophylla* CLas– Whole body | 12.88 | 4.14 | 53.68 | 471.04 | 33.3 | 356.48 | 1.19 | 350.74 | 19.36 | 51.44 | 135.21 | 4.17 |
| Nymph: *Citrus* spp. CLas– Whole body | 1.27 | 1.53 | 70.52 | 87.77 | 124.54 | 604.06 | 0.6 | 434.54 | 4.26 | 43.16 | 44.6 | 1.75 |
| Nymph: *Citrus* spp. CLas+ Whole body | 6.1 | 5.77 | 27.63 | 83.58 | 11 | 410.24 | 0.94 | 326.6 | 5 | 44.81 | 35.35 | 10.03 |
| Adult: *C. medica* CLas– Gut | 0.05 | 0.05 | 0.69 | 1.56 | 32.01 | 38.62 | 0 | 37.36 | 0 | 57.91 | 0.02 | 6.69 |
| Adult: *C. medica* CLas+ Gut | 0.07 | 0.03 | 1.25 | 0.57 | 39.99 | 30.13 | 0.03 | 37.35 | 0.02 | 72.09 | 0.03 | 6.49 |
| Adult: *C. medica* CLas+ High infection Whole body | 2.97 | 0.22 | 24.1 | 98.87 | 39.78 | 790.73 | 0.91 | 1123.64 | 6.69 | 38.29 | 20.34 | 8.69 |
| Adult: *C. medica* CLas+ Low infection Whole body | 2.98 | 0.28 | 33.97 | 104.33 | 30.33 | 735.91 | 1.16 | 846 | 6.79 | 40.08 | 19.07 | 11.88 |
| Adult: *C. medica* CLas– Whole body | 6.05 | 0.59 | 8.82 | 179.81 | 41.13 | 725.31 | 0.91 | 896.67 | 7.03 | 34.41 | 54.43 | 4.84 |
| Adult: *C. macrophylla* CLas– Whole body | 0.9 | 0.06 | 22.35 | 7.41 | 29.62 | 788.05 | 1.49 | 1098.32 | 1.34 | 49.79 | 2.32 | 13.07 |
| Adult: *Citrus* spp. CLas– Whole body | 0 | 0.09 | 11.7 | 2.99 | 88.19 | 533.99 | 1.42 | 368.82 | 0 | 42.11 | 0.86 | 6.88 |
| Adult: *Citrus* spp. CLas+ Whole body | 0 | 0.13 | 26.22 | 2.55 | 73.27 | 575.93 | 1.51 | 1055.86 | 0 | 44.96 | 0.51 | 9.02 |
| Adult: *Citrus* spp. CLas– midgut | 1.21 | 0.03 | 1.47 | 0.44 | 86.49 | 71.33 | 0 | 89.28 | 0.03 | 70.69 | 0.35 | 8 |
| Adult: *Citrus* spp. CLas+ midgut | 0.53 | 0.03 | 2.27 | 4.83 | 140.65 | 186 | 0.08 | 116.46 | 0.1 | 40.18 | 1.46 | 10.08 |
| Adult: *C. reticulata* CLas– Female abdomen | 0.46 | 0.16 | 7.75 | 1.32 | 81.55 | 946.67 | 0.48 | 883.26 | 0.13 | 29.65 | 0.35 | 21.83 |
| Adult: *C. reticulata* CLas– Female antennae | 0 | 0 | 32.8 | 18.89 | 85.16 | 1723.19 | 0 | 1750.55 | 0.22 | 23.29 | 0.64 | 4.25 |
| Adult: *C. reticulata* CLas– Female head | 0.21 | 0 | 32.76 | 10.97 | 181.03 | 1662.39 | 0.18 | 1915.3 | 0.25 | 19.52 | 0.43 | 3.33 |
| Adult: *C. reticulata* CLas– Female leg | 0.02 | 0 | 8.5 | 2.73 | 69.41 | 1315.12 | 0 | 2022.88 | 0.18 | 20.17 | 0.27 | 12.84 |
| Adult: *C. reticulata* CLas– Female terminal abdomen | 0.64 | 0 | 89.66 | 6.12 | 47.48 | 609.54 | 0.16 | 1068.81 | 0 | 13.73 | 1.61 | 17.82 |
| Adult: *C. reticulata* CLas– Female thorax | 0 | 0 | 15.82 | 0.96 | 131.37 | 1221.78 | 0 | 1482.48 | 0.58 | 19.88 | 0.91 | 6.94 |
| Adult: *C. reticulata* CLas– Male abdomen | 0.48 | 0.09 | 5.75 | 2.3 | 75.05 | 1139.68 | 0.26 | 1490.21 | 0.1 | 37.33 | 1.02 | 6.32 |
| Adult: *C. reticulata* CLas– Male antennae | 0.33 | 0 | 14.25 | 37.15 | 55.26 | 1689.26 | 0.16 | 1918.4 | 0.52 | 30.12 | 1.69 | 2.78 |
| Adult: *C. reticulata* CLas– Male head | 0 | 0 | 31.07 | 9.79 | 136.84 | 1761.42 | 0 | 1936.55 | 0.31 | 28.17 | 0.76 | 3.18 |
| Adult: *C. reticulata* CLas– Male leg | 0 | 0 | 14.41 | 1.03 | 77.63 | 1994.31 | 0.5 | 2939.9 | 0.16 | 27.21 | 0.73 | 5.36 |
| Adult: *C. reticulata* CLas– Male terminal abdomen | 1.83 | 0.02 | 13.6 | 8.5 | 24.59 | 823.39 | 0.37 | 1357.2 | 0.22 | 22.44 | 1.47 | 5.79 |
| Adult: *C. reticulata* CLas– Male thorax | 0 | 0 | 12.45 | 0.37 | 203.18 | 1386.57 | 0.03 | 1798.49 | 0.31 | 22.61 | 0.82 | 6.29 |
| Adult: *C. reticulata* CLas– Female antennae [28] | 0.53 | 0 | 25 | 8.71 | 151.75 | 1835.97 | 0.49 | 2699.83 | 0.57 | 29.29 | 0.65 | 8.61 |
| Adult: *C. reticulata* CLas– Female terminal abdomen [28] | 0.77 | 0 | 12.95 | 1.87 | 61.81 | 980.79 | 0.37 | 1114.55 | 0.07 | 23.81 | 0.6 | 16.12 |

| Table 5. (Continued) | | | | | | | | | | | | |
|---|---|---|---|---|---|---|---|---|---|---|---|---|
| **Gene ID** | **Cht5** | **Cht10-1** | **Cht10-2** | **Cht7** | **ChtPE** | **Idgf1** | **Idgf2** | **Idgf3** | **Cht6** | **Cht11** | **Cht3** | **ENGase** |
| Adult: *C. reticulata C*Las– Male antennae [28] | 0.44 | 0 | 23.52 | 20.19 | 140.14 | 2104.17 | 0.76 | 3582.55 | 1.36 | 37.96 | 1.61 | 5.23 |
| Adult: *C. reticulata C*Las– Male terminal abdomen [28] | 1.26 | 0.06 | 12.11 | 1.64 | 63.5 | 1132.34 | 0.98 | 1963.58 | 0 | 45.04 | 1.77 | 7.95 |

Expression values in transcripts per million (TPM) obtained from the Citrus Greening Expression Network [17] for annotated *Diaphorina citri* chitinase genes. Sample metadata including developmental stage, tissue, food source, and *C*Las exposure status are recorded in the first column. Cht: Chitinase; IDGF: Imaginal disc growth factor; ENGase: endo-B-N-acetylglucosaminidase.

## CONCLUSIONS

We have annotated 12 genes of the chitinase family from the citrus greening vector *D. citri*. We used BLAST, domain content and phylogenetic analysis to assign the predicted chitinase proteins into groups according to the current classification system [5]. *D. citri* has members of all chitinase groups except groups IV, VII, and IX (Table 4). We also determined that *D. citri* and several other sternorrhyncan hemipterans have a novel chitinase gene that appears to be the result of horizontal gene transfer.

## RE-USE POTENTIAL

Our curation of chitinase gene models and classification of chitinase proteins will be helpful to scientists wishing to carry out additional research on these genes. Chitinases are considered good targets for gene-based pest control methods, but research in other insects has shown that not all chitinases are essential. Our analysis will help researchers choose the best genes to target and will provide accurately annotated genes as a foundation for their work.

## DATA AVAILABILITY

The gene models are part of an updated official gene set (OGS) for *D. citri* submitted to NCBI under Bioproject PRJNA29447. Sequences of the annotated genes described here are available in the *GigaScience* GigaDB repository [34]. They are also included in an updated official gene set (OGS) linked to the same NCBI Bioproject. Genome assembly, transcriptome and official gene set sequences are currently available for BLAST and expression analysis on the Citrus Greening Solutions website [11].

## EDITOR'S NOTE

This article is one of a series of Data Releases crediting the outputs of a student-focused and community-driven manual annotation project curating gene models and, if required, correcting assembly anomalies, for *the Diaphorina citri* genome project [35].

## DECLARATIONS
## LIST OF ABBREVIATIONS

CBD: Chitin binding domain; CGEN: Citrus Greening Expression Network; Cht: Chitinase; *C*Las: *Candidatus* Liberibacter asiaticus; ENGase: endo-beta-N-acetylglucosaminidase; HGT: horizontal gene transfer; Idgf: Imaginal disc growth factor; MCOT: Maker, Cufflinks, Oases and Trinity; NCBI: National Center for Biotechnology Information; OGS: official gene set; RNA-seq: RNA sequencing; S/T: serine/threonine; TPM: transcripts per million.

## ETHICAL APPROVAL
Not applicable.

## CONSENT FOR PUBLICATION
Not applicable.

## COMPETING INTERESTS
The authors declare that they have no competing interests.

## FUNDING
This research was funded by USDA-NIFA grant 2015-70016-23028, HSI 1300394, 2020-70029-33199 and an Institutional Development Award (IDeA) from the National Institute of General Medical Sciences of the National Institutes of Health under grant number P20GM103418.

## AUTHORS' CONTRIBUTIONS
WBH, SJB, TD and LAM conceptualized the study; TD, SS, TDS and SJB supervised the study; SJB, TD, SS, and LAM contributed to project administration; SM, TDS, and BT conducted the investigation; PH, MF-G, and SS contributed to software development; SS, TDS, PH, and MF-G developed methodology; SJB, TD, WBH, and LAM acquired funding; TDS and SM prepared and wrote the original draft; SS, WBH and SJB reviewed and edited the draft.

## ACKNOWLEDGEMENT
We thank Dr. Josh Benoit for assistance with visualization of expression data.

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
