## [Reviewer Report]

Comments on revised manuscriptThe authors have revised the manuscript according to reviewer's comments. The current manuscript can be accepted.

---

## [Reviewer Report]

Reviewer name and names of any other individual's who aided in reviewer Mary Ann TuliDo you understand and agree to our policy of having open and named reviews, and having your review included with the published papers. (If no, please inform the editor that you cannot review this manuscript.)YesIs the language of sufficient quality?YesPlease add additional comments on language quality to clarify if needed
NAAre all data available and do they match the descriptions in the paper? YesAdditional CommentsAs with the other manuscripts, OGS v3 is mentioned, but this is not get available from the CGEN.
The data underlying table 4 and Fig 3 are available. Are the data and metadata consistent with relevant minimum information or reporting standards? See GigaDB checklists for examples <a href="http://gigadb.org/site/guide" target="_blank">http://gigadb.org/site/guide</a>YesAdditional CommentsIs the data acquisition clear, complete and methodologically sound?YesAdditional CommentsIs there sufficient detail in the methods and data-processing steps to allow reproduction?YesAdditional CommentsIs there sufficient data validation and statistical analyses of data quality? YesAdditional CommentsIs the validation suitable for this type of data?YesAdditional CommentsIs there sufficient information for others to reuse this dataset or integrate it with other data?YesAdditional CommentsAny Additional Overall Comments to the AuthorThis manuscript is a comprehensive description of the manual curation of the chitinase family genes, with clear aims and methodology.RecommendationAccept

---

## [Reviewer Report]

Reviewer name and names of any other individual's who aided in reviewer Hai-Zhong YuDo you understand and agree to our policy of having open and named reviews, and having your review included with the published papers. (If no, please inform the editor that you cannot review this manuscript.)YesIs the language of sufficient quality?YesPlease add additional comments on language quality to clarify if needed
Are all data available and do they match the descriptions in the paper? YesAdditional CommentsAre the data and metadata consistent with relevant minimum information or reporting standards? See GigaDB checklists for examples <a href="http://gigadb.org/site/guide" target="_blank">http://gigadb.org/site/guide</a>YesAdditional CommentsIs the data acquisition clear, complete and methodologically sound?NoAdditional CommentsIs there sufficient detail in the methods and data-processing steps to allow reproduction?NoAdditional CommentsIs there sufficient data validation and statistical analyses of data quality? YesAdditional CommentsIs the validation suitable for this type of data?NoAdditional CommentsIs there sufficient information for others to reuse this dataset or integrate it with other data?YesAdditional CommentsAny Additional Overall Comments to the AuthorThe manuscript presented by Shippy et al. revealed that chitinase family genes in Diaphorina citri. Chitin is widely distributed in nature and serves a variety of functions. In insects, chitin is a major structural component of the cuticle and peritrophic membrane, and plays an important role in molting; thus, chitin metabolism related genes can serve as a desired target for pest control. As described in background, chitinase plays an important role involved in digesting the polysaccharide polymer chitin. In the current study, the authors identified and annotated 12 chitinase family genes from D. citri and performed phylogenetic analysis. Additionally, the structural domains and expression patterns of D. citri chitinase genes were analyzed. In general, the manuscript can provide some useful information for D. citri control. This manuscript can be accepted after solving the following questions.
1. According to Table 1, 12 chitinases were identified, including CHT3, CHT5-7, CHT10-1, CHT10-2, CHT11, IDGF1-3, ENGase and CHTPE. However, CHT1-2, CHT4 and CHT8-9 seem to be missing. Please give a proper explain.
2. I suggested that the author should verify the expression levels of these chitinase genes by qPCR or Western blot. RecommendationMajor Revision